# Description and Comparative Genomics of *Algirhabdus cladophorae* gen. nov., sp. nov., a Novel Aerobic Anoxygenic Phototrophic Bacterial Epibiont Associated with the Green Alga *Cladophora stimpsonii*

**DOI:** 10.3390/life15030331

**Published:** 2025-02-21

**Authors:** Olga Nedashkovskaya, Sergey Baldaev, Alexander Ivaschenko, Evgenia Bystritskaya, Natalia Zhukova, Viacheslav Eremeev, Andrey Kukhlevskiy, Valeria Kurilenko, Marina Isaeva

**Affiliations:** 1G.B. Elyakov Pacific Institute of Bioorganic Chemistry, Far Eastern Branch, Russian Academy of Sciences, Pr. 100 Let Vladivostoku 159, 690022 Vladivostok, Russia; baldaevsergey@gmail.com (S.B.); ep.bystritskaya@yandex.ru (E.B.); wieremeew@gmail.com (V.E.); valerievk141075@gmail.com (V.K.); 2Institute of High Technology and Advanced Materials, Far Eastern Federal University, Ajax Bay 10, Russky Island, 690922 Vladivostok, Russia; sashann2003@gmail.com; 3A.V. Zhirmunsky National Scientific Center of Marine Biology, Far Eastern Branch, Russian Academy of Sciences, Palchevskogo Street 17, 690041 Vladivostok, Russia; nzhukova35@list.ru (N.Z.); ad.kukhlevskiy@gmail.com (A.K.)

**Keywords:** marine bacteria *Algirhabdus cladophorae*, *Roseobacteraceae*, green alga *Cladophora stimpsonii*, whole-genome sequence, pan-genome and singleton analysis

## Abstract

A novel, strictly aerobic, non-motile, and pink-pigmented bacterium, designated 7Alg 153^T^, was isolated from the Pacific green alga *Cladophora stimpsonii*. Strain 7Alg 153^T^ was able to grow at 4–32 °C in the presence of 1.5–4% NaCl and hydrolyze L-tyrosine, gelatin, aesculin, Tweens 20, 40, and 80 and urea, as well as produce catalase, oxidase, and nitrate reductase. The novel strain 7Alg 153^T^ showed the highest similarity of 96.75% with *Pseudaestuariivita rosea* H15^T^, followed by *Thalassobius litorarius* MME-075^T^ (96.60%), *Thalassobius mangrovi* GS-10^T^ (96.53%), *Tritonibacter litoralis* SM1979^T^ (96.45%), and *Marivita cryptomonadis* CL-SK44^T^ (96.38%), indicating that it belongs to the family *Roseobacteraceae*, the order *Rhodobacteales*, the class *Alphaproteobacteria*, and the phylum *Pseudomonadota*. The respiratory ubiquinone was Q-10. The main polar lipids were phosphatidylethanolamine, phosphatidylglycerol, diphosphatidylglycerol, phosphatidylcholine, two unidentified aminolipids, and one unidentified lipid. The predominant cellular fatty acids (>5%) were C_18:1_ ω7c, C_16:0_, C_18:0_, and 11-methyl C_18:1_ ω7c. The 7Alg 153^T^ genome is composed of a single circular chromosome of 3,786,800 bp and two circular plasmids of 53,157 bp and 37,459 bp, respectively. Pan-genome analysis showed that the 7Alg 153^T^ genome contains 33 genus-specific clusters spanning 92 genes. The COG20-annotated singletons were more often related to signal transduction mechanisms, cell membrane biogenesis, transcription, and transport, and the metabolism of amino acids. The complete photosynthetic gene cluster (PGC) for aerobic anoxygenic photosynthesis (AAP) was found on a 53 kb plasmid. Based on the phylogenetic evidence and phenotypic and chemotaxonomic characteristics, the novel isolate represents a novel genus and species within the family *Roseobacteraceae*, for which the name *Algirhabdus cladophorae* gen. nov., sp. nov. is proposed. The type strain is 7Alg 153^T^ (=KCTC 72606^T^ = KMM 6494^T^).

## 1. Introduction

The green algae or *Chlorophyta* widely distributed in the aquatic and terrestrial habitats are an important part of the marine ecosystem. The most *Chlorophyta* members are unicellular microorganisms which have recently been intensively studied as a valuable source of biologically active substances such as β-carotene and lipids, including unsaturated fatty acids, proteins, and vitamins [1,2]. The green marine macroalgae (green seaweeds) generally belong to the class *Ulvophyceae*, which is mainly known by several representative genera as *Ulva* and *Cladophora*. Some of them have shown significant biotechnological potential, in particular due to their bioactive lipids and unique polysaccharide composition. The genus *Ulva* members contain a sulfated polysaccharide ulvan, which is characterized by immunomodulatory, antimicrobial, and anticoagulant activities [3]. A sulfated polysaccharide, isolated from the marine macroalga *Cladophora oligoclada*, is also considered to become an anticoagulant agent for prevention and therapy of thrombotic diseases [4]. The other marine macroalga *Cladophora stimpsonii* was shown to contain high amounts of α-linolenic acid in addition to linoleic acid, which is characteristic of most *Ulvales* species [5]. Moreover, the pharmaceutical utility of the genus *Cladophora* has recently been summarized [6]. Members of this group have been shown to have antioxidant, antidiabetic, antihypertensive, antiparasitic, antimicrobial, anticancer, and cytotoxic effects.

In addition to valuable biological activities, the green algae provide a habitat for diverse epiphytic bacteria, the cultural isolation of which allows for a better understanding of their ecological role in the conservation and management of the algae host. Many novel bacterial taxa belonging to the phyla *Pseudomonadota*, *Actinomycetota*, *Bacteroidota*, and *Verrucomicrobiota* were recently isolated and described, including alphaproteobacteria *Litorimonas cladophorae*, *Amylibacter ulvae*, *Sulfitobacter algicola*, and *Hoeflea ulvae*; gammaproteobacterium *Marinomonas algicola*; actinobacterium *Nocardiopsis codii*; and representatives of the classes *Flavobacteriia* and *Opitutia* such as *Polaribacter reichenbachii*, *Kordia ulvae*, *Hyunsoonleella ulvae*, *Winogradskyella ulvae*, and *Winogradskyella algicola* and *Coraliomargarita algicola*, respectively [7,8,9,10,11,12,13,14,15,16,17,18].

The family *Roseobacteraceae* was proposed by Liang et al. in 2021 for representatives of the *Roseobacter* clade of the family *Rhodobacteraceae* based on whole-genome analyses with the type genus *Roseobacter* [19]. This family currently includes 136 genera with validly published names (https://lpsn.dsmz.de/family/roseobacteraceae (accessed on 23 December 2024) [20]). Some members of the *Roseobacter* clade are known as aerobic anoxygenic phototrophic bacteria, capable of using both light energy and organic carbon sources. The genes responsible for aerobic anoxygenic photosynthesis are found both on chromosomes and on plasmids [21,22,23].

Previously, during the study of bacterial communities associated with the green alga *Cladophora stimpsonii*, a common inhabitant of the Sea of Japan, *L. cladophorae* sp. nov., a new alphaproteobacterium [9], and *Lacinutrix cladophorae* sp. nov., a new a flavobacterium [10], were isolated.

In course of the study of the diversity of bacterial epibionts recovering from the green alga *Cladophora stimpsonii*, the pink-pigmented alphaproteobacterium, designated 7Alg 153^T^, was isolated and investigated using a polyphasic approach. According to phenotypic, chemotaxonomic, genotypic, and phylogenomic data, we proposed to classify strain 7Alg 153^T^ to a new genus and species of the family *Roseobacteraceae*.

## 2. Materials and Methods

### 2.1. Bacterial Isolation and Maintenance

Strain 7Alg 153^T^ was isolated from the green alga *Cladophora stimpsonii* collected in Troitsa Bay, Gulf of Peter the Great, the Sea of Japan (42°38′55″ N; 131°06′47″ E) in September 2009, as described previously by a standard dilution plating method [9]. The algal sample was washed with sterilized seawater, and a piece of algal fronds (5 g) was aseptically transferred to a glass homogenizer and homogenized in 10 mL of sterile seawater. The resulting homogenate (0.1 mL) was distributed onto marine agar 2216 (MA, Difco, Sparks, MD, USA) plates. The strain was obtained from a single colony after incubation at 28 °C for 7 days and subsequently purified on the same medium. After isolation, the strain was placed in Marine Broth (MB, BD Difco^TM^, Port-de-Claix, France) with 20% (*v*/*v*) glycerol for storage at −80 °C.

### 2.2. Phenotypic and Chemotaxonomic Characterization

The phenotype characteristics of strain 7Alg 153^T^ were examined at 28 °C using the commercial kits API 20E, API 50CH, API ID 32GN, API 20NE, and API ZYM (bioMérieux, Marcy l’Etoile, France), as recommended by the manufacturer. The morphology of cells grown on MA for 48 h at 28 °C was studied by a transmission electron microscope Libra 120 FE (Carl Zeiss, Oberkochen, Germany) provided by the A.V. Zhirmunsky National Scientific Center of Marine Biology, the Far Eastern Branch of the Russian Academy of Sciences. Gram staining was performed according to the ASM protocol [24]. Catalase and oxidase activities, fermentative or oxidative carbohydrate utilization, as well as DNA, polysaccharide, protein, and urea degradation, Tweens 20, 40, and 80 hydrolysis, nitrate reduction, hydrogen sulfide production, and susceptibility to antibiotics were tested as previously described [9,25]. The temperature, pH, and salinity ranges for the growth of strain 7Alg 153^T^ were measured at 4–42 °C, pH 5–11, and NaCl concentrations of 0–10.0% (*w*/*v*), as described [25].

Fatty acid methyl esters and polar lipids of strain 7Alg 153^T^ were extracted and analyzed as described previously [26], using cells grown on MA at 28 °C for 48 h. Isoprenoid quinones were isolated with a chloroform/methanol extraction system and purified by TLC using a n-hexane and diethyl ether mixture. Isoprenoid quinone composition was determined by HPLC on the Shimadzu LC-10A (Shimadzu Corporation, Kyoto, Japan) using a Supelcosil LC-18 column (Supelco, Bellefonte, PA, USA) and acetonitrile/2-propanol mobile phase at a 0.5 mL/min flow rate as described previously [27]. Quinone detection was monitored at 275 nm.

For a polyphasic taxonomic study, data on *Pseudaestuariivita rosea* H15^T^ [28], *Planktomarina temperata* RCA23^T^ [29], *Litoreibacter albidus* KMM 3851^T^ [30], *Nereida ignava* DSM 16309^T^ [31], and *Sulfitobacter algicola* 1151^T^ [13], which were neighbors of 7Alg 153^T^ on the genomic tree, were added in all tables as reference strains.

### 2.3. 16S rRNA and RpoC Sequence and Phylogenetic Analysis

The 16S rRNA gene amplification of the 7Alg 153^T^ strain was carried out with primers 27F (5′-AGAGTTTGATCMTGGCTCAG-3′) and 1492R (5′-TACGGTTACCTTGTTACGACTT-3′) [32] and sequenced on a SeqStudio^TM^ Genetic Analyzer (ThermoFisher Scientific, Singapore). The similarities of 16S rRNA genes between strains were calculated on the EzBioCloud server [33], and the phylogenetic relationships were estimated using MEGA X version 10.2.1 [34] and the GGDC web server (http://ggdc.dsmz.de/, accessed on 17 October 2024) [35], using the DSMZ pipeline for single genes [36]. A neighbor-joining (NJ) tree was obtained based on the K2P model. Maximum likelihood (ML) and maximum parsimony (MP) trees were inferred with RAxML [37] and TNT [38]. The ML tree was obtained based on the GTR+GAMMA model. All trees were tested by bootstrapping with 1000 replicates.

### 2.4. Whole-Genome Sequencing and Genome-Based Phylogenetic Analysis

The genomic DNA of 7Alg 153^T^ was extracted by the NucleoSpin Tissue kit (Macherey-Nagel, Düren, Germany). The quality and quantity of the genomic DNA was estimated by agarose gel electrophoresis and the Qubit 4.0 Fluorometer (Thermo Fisher Scientific, Waltham, MA, USA). The DNA sequencing library obtained with a Nextera DNA Flex kit (Illumina, San Diego, CA, USA) was sequenced using 2 × 150 bp paired-end runs on an Illumina MiSeq platform. The nanopore DNA library was prepared with EXP-NBD104 and SQK-LSK109 kits (ONT, Oxford, UK). The FASTQ data preprocessing and quality control were performed using Trimmomatic version 0.39 [39] and FastQC version 0.11.8 (https://www.bioinformatics.babraham.ac.uk/projects/fastqc/, accessed on 27 May 2024), respectively. Dorado version 0.4.3 (ONT, Oxford, UK) was used to qualitatively filter Nanopore reads, removing sequences shorter than 1000 bp. The hybrid assembly of reads was performed using Unicycler version 0.4.8 [40]. Samtools version 1.3 was used to estimate sequencing depth [41]. The completeness and contamination values were obtained by CheckM version 1.1.3 using a taxonomic-specific workflow (family *Rhodobacteriaceae*) [42]. The complete genome was annotated using PGAP, RAST, and EggNOG [43,44,45]. The overall genomic relatedness indices (OGRIs) such as ANI (Average Nucleotide Identity), AAI (Average Amino Acid Identity), and dDDH (in silico DNA-DNA hybridization) were calculated on the online server ANI/AAI-Matrix [46] and TYGS platform [47], respectively.

Anvi’o version 8 was used for metabolism estimation and pan-genomic analysis [48]. The pan-genome reconstruction was conducted using the anvi’o workflow, as indicated at https://merenlab.org/2016/11/08/pangenomics-v2/ (accessed on 20 November 2024).

To extract additional functional and ecological features, the Protologger (Galaxy Version 1.0.0) [49] was utilized. Biosynthetic gene cluster annotation for secondary metabolite biosynthesis was performed using antiSMASH server version 7.0.0 (https://antismash.secondarymetabolites.org, accessed on 12 November 2024) [50].

Orthologous gene clusters and singletons were analyzed using OrthoVenn3 (https://orthovenn3.bioinfotoolkits.net/home, accessed on 18 June 2024) [51]. The orthologous clusters were identified with the default parameters, with a 1 × 10^−5^ e-value cutoff for all protein similarity comparisons and 1.5 inflation value for the generation of orthologous clusters.

The functional and ecological analyses of the strain were performed using the Protologger web tool [49], at http://www.protologger.de/ accessed on 20 October 2024. In Protologger, the 16S rRNA gene sequence was matched against a database containing 19,000 amplicon datasets, and the genome was compared to 49,094 high-quality metagenome-assembled genomes (MAGs).

## 3. Results

### 3.1. Phylogenetic and Phylogenomic Analyses

To carry out the phylogenetic position of strain 7Alg 153^T^, we amplified its 16S rRNA gene sequence of 1384 bp. Pairwise values of the 16S rRNA sequence similarity were obtained using EzBioCloud server (https://www.ezbiocloud.net/ (accessed on 21 November 2024) [33]. From this analysis, strain 7Alg 153^T^ showed the highest similarity of 96.75% with *Pseudaestuariivita rosea* H15^T^, followed by *Thalassobius litorarius* MME-075^T^ (96.60%), *Thalassobius mangrovi* GS-10^T^ (96.53%), *Tritonibacter litoralis* SM1979^T^ (96.45%), and *Marivita cryptomonadis* CL-SK44^T^ (96.38%). However, the 16S rRNA phylogenetic trees reconstructed with NJ, ML, and MP algorithms (Figure 1) demonstrated very low-support branches (54%) and discordant topologies.

To select genomes of type strains of closely related genera for phylogenomic analysis, the RpoC protein sequence of 7Alg 153^T^ was used as a query in a Blast search in the NCBI database. The RpoC protein is a DNA-directed RNA polymerase, the subunit beta’ [EC:2.7.7.6]. Sometimes, the RpoC protein sequence is used as a phylogenetic marker to determine the approximate phylogenetic position of new *Roseobacteraceae* and *Paracoccaceae* strains [25,52,53]. An NJ phylogenetic tree constructed based on 36 RpoC protein sequences of type strains of *Roseobacteraceae* and *Paracoccacea* genera showed that 7Alg 153^T^ clusters with *Nereida ignava* DSM 16309^T^ [31] and *Planktomarina temperata* RCA23^T^ [29] with very high support (Appendix A). Three strains of *Pseudaestuariivita rosea* H15^T^, *Sulfitobacter algicola* 1151^T^, and *Tritonibacter litoralis* SM1979^T^ are not type strains of the corresponding genera. Since the first two strains formed a clade close to the clade of 7Alg 153^T^, their genomes were taken for further analysis. The latter strain was replaced by the type species strain of the genus *Tritonibacter* due to its significantly distant position.

Then, to accurately determine the taxonomic position of 7Alg 153^T^, a phylogenomic tree was reconstructed based on the concatenated sequences of 400 translated proteins extracted from 21 selected genomes of type species strains of closely related *Roseobacteraceae* genera, including *Pseudaestuariivita rosea* H15^T^ and *Sulfitobacter algicola* 1151^T^. The resulting phylogenomic tree showed that 7Alg 153^T^ forms a distinct lineage at the genus level (Figure 2), clustering with type species of the genera *Nereida*, *Planktomarina,* and *Litoreibacter*. In addition, *Pseudaestuariivita rosea* H15^T^ and *Sulfitobacter algicola* 1151^T^ retained their positions as the closest neighbors of 7Alg 153^T^ in this genomic tree, as it was determined by the RpoC protein tree (Appendix A). Recently, the taxonomic status of the genus *Sulfitobacter* was revised based on genome analysis, identifying *Sulfitobacter algicola* 1151^T^ as a type species of the new genus *Parasulfitobacter* gen. nov. [54]. Given the position of *Pseudaestuariivita rosea* H15^T^ on the tree, we can confidently state that this species does not belong to the genus *Pseudaestuariivita* and should be reclassified into a new genus of the family *Roseobacteraceae*.

The 16S rRNA gene sequence similarity values between 7Alg 153^T^ and closely related type strains were as follows: *Pseudaestuariivita rosea* H15^T^ (96.75%), *Sulfitobacter algicola* 1151^T^ (95.24%), *Planktomarina temperata* RCA23^T^ (95.25%), *Nereida ignava* 2SM4^T^ (94.95%), and *Litoreibacter albidus* KMM 3851^T^ (94.55%). The OGRI values between 7Alg 153^T^ and type species strains of the genera *Nereida*, *Planktomarina*, and *Litoreibacter*, as well as and *Pseudaestuariivita rosea* H15^T^ and *Sulfitobacter algicola* 1151^T^ (representatives of potential novel genera), were 74–76% for ANI (Appendix A), 63–66% for AAI (Appendix A), and lower than 19.8% for dDDH (formula d4).

Considering the AAI values of 45–65% proposed for outlining bacterial genera [55] and the new strain phylogenomic position indicating that it does not belong to any of the known genera, the new strain 7Alg 153^T^ should be classified as a representative of a new genus and a new species of the family *Roseobacteraceae*.

### 3.2. Genome Analysis of Genus-Related Features

The final de novo genome assembly of 7Alg 153^T^ resulted in a single circular chromosome of 3,786,800 bp in size and two circular plasmids of 53,157 bp and 37,459 bp, respectively (Figure 3). The overall GC content of 7Alg 153^T^ genome was 55%. Based on NCBI PGAP [44], the genome contains a total of 3761 genes, including 3703 protein-coding genes, 38 tRNAs, and three rRNAs (one *rrn* operon). The chromosome has the *dnaA* origin of replication determined by Ori-Finder 2 [56], which was confirmed with a GC skew plot (Figure 3).

The 16S rRNA gene sequence extracted from the genome was 100% identical to the PCR-amplified one (PQ521906.1). Based on phylogenomic tree results, type species of the genera *Nereida*, *Planktomarina*, and *Litoreibacter*, as well as *Pseudaestuariivita rosea* H15^T^ and *Sulfitobacter algicola* 1151^T^, were taken for comparative genomic analysis. The genomic features of 7Alg 153^T^ and other type strains from closely related genera are shown in Table 1. The observed genome characteristics correspond well to the updated proposed minimal standards for prokaryotic taxonomy [58,59].

To clarify genus-related features, a comparative genomic analysis of orthologous genes was carried out with OrthoVenn 3 v3.0 [51]. The analysis revealed 4136 orthologous gene clusters; among them, 1373 were single-copy clusters (Figure 4). The core genome was composed of 1452 orthologous clusters, while the accessory genome was composed of 2763 ones. The total number of unique genes (singletons) was 4951. The majority of 7Alg 153^T^ clusters was shared with those of *Planktomarina temperata* RCA23^T^ (97) and *Litoreibacter albidus* DSM 26922^T^ (82) (Figure 4). The maximum number of gene clusters was shared between *Pseudaestuariivita rosea* H15^T^ and *Sulfitobacter algicola* 1151^T^, which is well supported by their position on the genomic tree (Figure 2). The 7Alg 153^T^ genome contained 33 genus-specific clusters spanning 92 genes. The range of other genus-specific clusters was from 14 (35 genes) for *Litoreibacter albidus* DSM 26922^T^ to 59 (153 genes) for *Planktomarina temperata* RCA23^T^.

In addition, metabolic pathway reconstruction was performed with Anvi‘o pangenomic workflow platform [48]. A total of 10,762 gene clusters with 22,120 gene calls were identified (Figure 5a). The clusters were split into core (1194 gene clusters, 7362 gene calls), shell (713 gene clusters, 3287 gene calls), cloud (1684 gene clusters, 4077 gene calls), and singletons (7171 gene clusters, 7394 gene calls). According to the COG20 category annotation of the singletons (Appendix A), the most abundant functional categories were signal transduction (T), general function (R), transcription (K), membrane biogenesis (M), and transport and metabolism of amino acids (E).

Some significant differences in metabolic pathway completeness were revealed between the new and type strains (Figure 5b). Only 7Alg 153^T^ showed the presence of complete M00009 (citrate cycle) and M00854 (glycogen biosynthesis). In addition, M00011 (citrate cycle, second carbon oxidation) was predicted for both 7Alg 153^T^ and *Planktomarina temperata* RCA23^T^. Modules M00156 (cytochrome c oxidase) and M00091 (phosphatidylcholine biosynthesis) were absent in 7Alg 153^T^ and *Planktomarina temperata* RCA23^T^, while the latter was also not found in *Nereida ignava* DSM 16309^T^. All strains except for *Litoreibacter albidus* DSM 26922^T^ possessed M00597 (anoxygenic photosystem II).

Since the photosynthetic system was detected in all five genomes, its detailed analysis was performed (Figure 6). The complete photosynthetic gene cluster (PGC) for aerobic anoxygenic photosynthesis (AAP) was found on the plasmid of 7Alg 153^T^, designated as pACLA_53 (Figure 3). This plasmid harbors key genes of bacteriochlorophyll A biosynthesis (*bch*), proteins forming the reaction centers (*puf*), assembly factors (*puh*), and carotenoid biosynthesis (*crt*) [21]. Notably, the presence of a *dnaA*-like gene on pACLA_53 may suggest the stable maintenance of the AAP system in 7Alg 153^T^ [60]. Furthermore, two PGC regions were suggested arise through horizontal gene transfer (HGT) events, based on the Alien Hunter results [61]. The 7Alg 153^T^ PGC showed similarity in both sequences and cluster architecture to the PGCs of *Nereida ignava* DSM 16309^T^, *Planktomarina temperata* RCA23^T^, *Sulfitobacter algicola* 1151^T^, and *Pseudaestuariivita rosea* H15^T^ (Figure 6a). This suggests that the presence of the pACLA_53 plasmid may be part of an adaptation strategy of 7Alg 153^T^, allowing for its survival in nutrient-poor environmental conditions, as described for *Nereida ignava* DSM 16309^T^ and *Planktomarina temperata* RCA23^T^ [22,23].

Regarding other methods of obtaining additional energy described for the *Roseobacter* group [62], only *Planktomarina temperata* RCA23^T^ encodes both forms of CO dehydrogenase (CODH) required for CO oxidation, while the others encode only form II (*coxSLM*, CODH II) and are thus predicted to be unable to oxidize CO. The sox gene cluster (*soxAXYZBCD*), encoding a Sox multienzyme system for sulfur oxidation, was present in all genomes studied. All genomes lacked RuBisCo genes, which allow for CO_2_ assimilation for survival in autotrophic conditions.

The presence of biosynthesis gene clusters (BGCs) in the 7Alg 153^T^ genome, responsible for the synthesis of secondary metabolites, was studied using the antiSMASH server [50]. This genome was found to contain three gene clusters encoding the terpene, hserlactone, and RiPP-like type metabolites. The terpene gene cluster showed 100% similarity to the carotenoid BGS. The functional and ecological characteristics of 7Alg 153^T^ were studied by Protologger software [50]. Among the 3740 coding sequences, 206 were annotated as transport proteins and 37 were assumed to be responsible for secretion. Anoxygenic photosystem II (*pufL*, *pufM*) was detected in the 7Alg 153^T^ genome. It was predicted that it can utilize starch as a carbon source and urea (EC:3.5.1.5). It is assumed that 7Alg 153^T^ can produce L-cysteine and acetate from sulfide and L-serine (EC:2.3.1.30, 2.5.1.47), as well as L-glutamate from ammonia via L-glutamine (EC:6.3.1.2, 1.4.1.-) and synthesized cobalamin (EC:2.5.1.17, 6.3.5.10, 6.2.1.10, 2.7.1.156). Flagellar protein genes detected in the 7Alg 153^T^ genome were 2 *flh* (*flhA* and *flhB*), 12 *flg* (*flgA*, *flgB*, *flgC*, *flgD*, *flgE*, *flgF*, *flgG*, *flgH*, *flgI*, *flgJ*, *flgK*, and *flgL*), 11 *fli* (*fliC*, *fliE*, *fliF*, *fliG*, *fliI*, *fliK*, *fliL*, *fliN*, *fliO*, *fliQ*, and *fliR*), and 3 *mot* (*motA*, *motB*, and *motE*). A total of 174 carbohydrate-active enzymes (CAZymes) were predicted in the genome, mainly glycoside hydrolases (55 GHs).

Comparative analysis did not reveal any MAGs matching the 7Alg 153^T^ genome. Based on 16S rRNA amplicon database OTUs, the gene sequence of 7Alg 153^T^ was represented in coral metagenomes (43.7% of samples), followed by marine (22.0% of samples) and marine sediment (14.3%) metagenomes, consistent with its isolation source (Appendix A).

### 3.3. Morphological, Physiological, and Biochemical Characteristics

The detailed physiological and biochemical characteristics of strain 7Alg 153^T^ are given in the genus and species description, Appendix A and Table 2 and Table 3. Cells of strain 7Alg 153^T^ were Gram stain-negative, strictly aerobic, and non-motile rods (0.3–0.6 μm wide and 0.9–2.3 μm long), which can form pink-pigmented colonies on MA. Strain 7Alg 153^T^ was susceptible to ampicillin, benzylpenicillin, carbenicillin, cefalexin, cefazolin, chloramphenicol, erythromycin, doxycycline, gentamicin, kanamycin, lincomycin, nalidixic acid, neomycin, ofloxacin, oleandomycin, oxacillin, rifampicin, streptomycin, tetracycline, and vancomycin, and it was resistant to polymyxin. The G+C content from the 7Alg 153^T^ genome was 55 mol% (Table 1). The isolate shared many phenotypic properties with the closest relatives, including the respiratory type of metabolism and oxidase production and the absence of agarase and amylase activities. However, strain 7Alg 153^T^ differed from all the strains studied by the presence of nitrate reductase activity. Moreover, the sets of phenotypic features shown in Table 2 clearly distinguished 7Alg 153^T^ from its nearest neighbors.

### 3.4. Chemotaxonomic Characteristics

The predominant fatty acids (>5% of the total fatty acids) of strain 7Alg 153^T^ were C_18:1_ ω7c (70.2%), C_16:0_ (7.5%), C_18:0_ (7.5%), and 11-methyl C_18:1_ ω7c (5.1%) (Table 3). The fatty acid composition of the strain studied was similar to those of the reference strains, although sufficient differences were found in the proportions of some fatty acids (Table 3). Thus, a significant amount of fatty acid C_18:1_ ω7c found in strain 7Alg 153^T^ was absent in *Pseudaestuariivita rosea* H15^T^ and *Litoreibacter albidus* KMM 3851^T^. At the same time, the small amount of fatty acid C_20:1_ ω7c in the novel strain in comparison with *Pseudaestuariivita rosea* H15^T^, *Sulfitobacter algicola* 1151^T^, and *Litoreibacter albidus* KMM 3851^T^ can be a helpful feature for their differentiation. The main polar lipids found in strain 7Alg 153^T^ included phosphatidylethanolamine, phosphatidylglycerol, diphosphatidylglycerol, phosphatidylcholine, two unidentified aminolipids, and one unidentified lipid (Appendix A). The novel isolate can be distinguished from its closest relatives by the presence of diphosphatidylglycerol [28,29,30,32]. In contrast with *Litoreibacter albidus* KMM 3851^T^, *Sulfitobacter algicola* 1151^T^, and *Pseudaestuariivita rosea* H15^T^, strain 7Alg 153^T^ contained aminolipids [13,28,30]. The presence of phospholipids and the absence of phosphatidylethanolamine and phosphatidylcholine clearly separated the algal isolate from *Planktomarina temperata* RCA23^T^ [29].

## 4. Conclusions


**Description of *Algirhabdus* gen. nov.**


*Algirhabdus* (Al.gi.rhab’dus. L. fem. n. alga, alga or seaweed; Gr. fem. n. rhabdos, a rod; N.L. fem. n. *Algirhabdus*, referring to a rod-shaped bacterium from alga (associated with algae).

Cells are Gram stain-negative, strictly aerobic, rod-shaped, non-spore-forming, and non-motile, as well as catalase- and oxidase-positive. The dominant fatty acids (>5%) are C_18:1_ ω7c, C_16:0_, C_18:0_, and 11-methyl C_18:1_ ω7c. The polar lipids are phosphatidylethanolamine, phosphatidylglycerol, diphosphatidylglycerol, phosphatidylcholine, unidentified aminolipids, and an unidentified lipid. The major respiratory quinone is Q-10. Phylogenetically, the genus belongs to the family *Roseobacteraceae*, the order *Rhodobacterales*, the class *Alphaproteobacteria*, and the phylum *Pseudomonadota*. The type species is *Algirhabdus cladophorae*.


**Description of *Algirhabdus cladophorae* sp. nov.**


*Algirhabdus cladophorae* (cla.do.pho’rae. N.L. gen. fem. n. cladophorae, of/from *Cladophora*, the genus name *Cladophora stimpsonii*, the host of the organism).

Cells are heterotrophic, strictly aerobic, non-motile, Gram stain-negative rods, and they are 0.3–0.6 μm wide and 0.9–2.3 μm long. On marine agar, colonies are circular, shiny, mucous, slightly convex, with entire edges, 1–2 mm in diameter, and pink-colored. Growth occurs at 4–32 °C (optimum is 25–28 °C) and pH 6.0–9.5 (optimum is 7.0–8.0), as well as with 1.5–4% NaCl (optimum is 2–3% NaCl). Seawater or artificial seawater is required for growth. Catalase and oxidase activities are present. Aesculin, gelatin, L-tyrosine, urea, and Tweens 20, 40, and 80 are hydrolyzed but agar, casein, starch, DNA, chitin, and CM–cellulose are not. Acid is produced from D- arabinose, L-arabinose, D-fructose, D-galactose, D-glucose, lyxose, maltose, melibiose, L-rhamnose, sucrose, trehalose, D-xylose, turanose, amygdalin, sorbitol, and arbuthin but not from D-cellobiose, D-fucose, D-lactose, mannose, raffinose, ribose, N-acetylglucosamine, mannitol, glycerol, and citrate. D-lactose, maltose, L-rhamnose, N-acetyl-glucosamine, erythritol, glycerol, sorbitol, acetate, citrate, L-alanine, L-glutamate, L-histidine, L-threonine, glucuronate, gluconate, and propionate are utilized. In the API 20NE gallery, it is positive for nitrate reductase, the hydrolysis of gelatin, urea, and PNPG, and the assimilation of maltose and gluconate. In the API 20E kit, ONPG, citrate, urease, gelatinase, glucose, sorbitol, L-rhamnose, sucrose, melibiose, amygdalin, and L-arabinose tests were positive. With API 50 CH strips, positive results were obtained for the oxidation of glycerol, erythritol, D-arabinose, L-arabinose, D-xylose, galactose, glucose, fructose, amygdalin, arbuthin, aesculin, maltose, sucrose, trehalose, melizitose, turanose, and lyxose. In the API ID 32GN kit, N-acetylglucosamine, D-ribose, D-sucrose, D-maltose, itaconic acid, subaric acid, sodium acetate, lactic acid, 3-hydrobenzoic acid, D-glucose, salicin, D-melibiose, L-fucose, D-sorbitol, L-arabinose, propionic acid, capric acid, valeric acid, sodium citrate, L-histidine, potassium 2-keto-gluconate, 3-hydroxybutiric acid, and L-prolin are assimilated. According to the API ZYM tests, alkaline phosphatase, esterase (C4), esterase lipase (C8), leucine arylamidase, and naphtol-AS-BI-phosphohydrolase activities are present but lipase (C14), cystine arylamidase, trypsin, α-chymotrypsin, α-galactosidase, β-galactosidase, β-glucuronidase, α-glucosidase, β-glucosidase, N-acetyl-β-glucosaminidase, α-mannosidase, and α-fucosidase activities are absent. Acid phosphatase activity is weakly positive. Nitrate is reduced. Hydrogen sulfide, indole, and acetoin are not produced. The prevalent fatty acids (>5%) are C_18:1_ ω7c, C_16:0_, C_18:0_, and 11-methyl C_18:1_ ω7c. The polar lipids are phosphatidylethanolamine, phosphatidylglycerol, diphosphatidylglycerol, phosphatidylcholine, two unidentified aminolipids, and one unidentified lipid. The DNA G+C content of strain 7Alg 153^T^ is 55 mol%.

The DDBJ/GenBank accession numbers for the 16S rRNA gene and genome sequences of strain 7Alg 153^T^ are PQ521906 and GCA_044772865.1, respectively.

The type strain 7Alg 153^T^ (=KCTC 72606^T^ = KMM 6494^T^) was isolated from the green alga *Cladophora stimpsonii* collected from Troitsa Bay, Gulf of Peter the Great, the Sea of Japan, Russia.

## Figures and Tables

**Figure 1 life-15-00331-f001:**
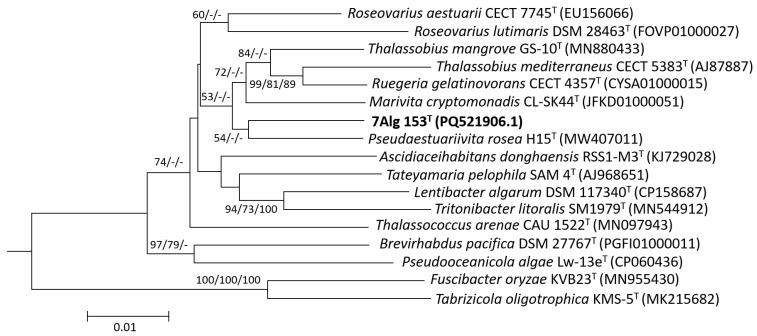
16S rRNA phylogenetic tree of 7Alg 153^T^ (in bold) and type strains of *Roseobacteracea* species. The branch lengths represent the expected number of substitutions per site. The bootstrap values larger than 60% are shown above the branches as NJ/ML/MP numbers based on 1000 replicates. The scale bar is 0.01 substitutions per nucleotide position.

**Figure 2 life-15-00331-f002:**
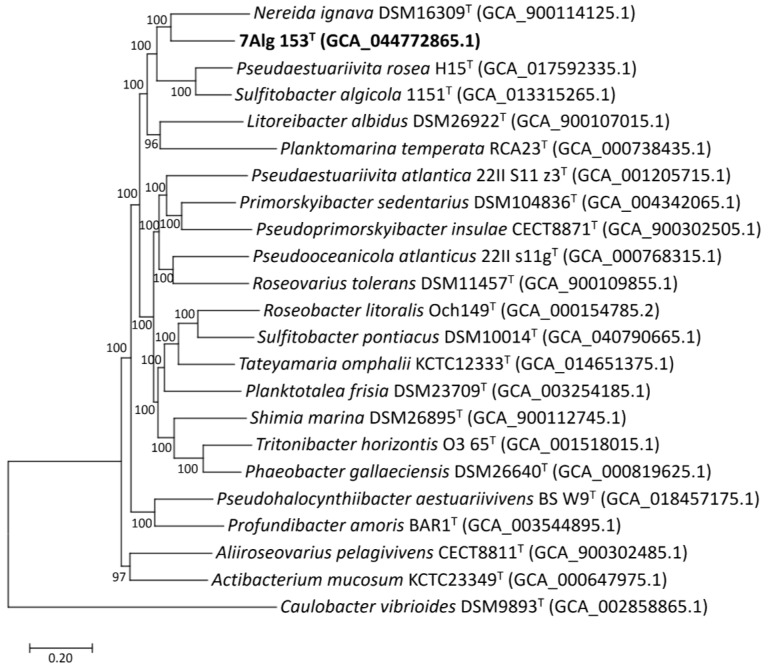
ML phylogenomic tree showing position of 7Alg153^T^ among type strains of *Roseobacteraceae* genera based on concatenated sequences of 400 translated proteins. Bootstrap values are based on 100 replicates. *Caulobacter vibrioides* was used as outgroup. Scale bar is 0.20 substitutions per amino acid position.

**Figure 3 life-15-00331-f003:**
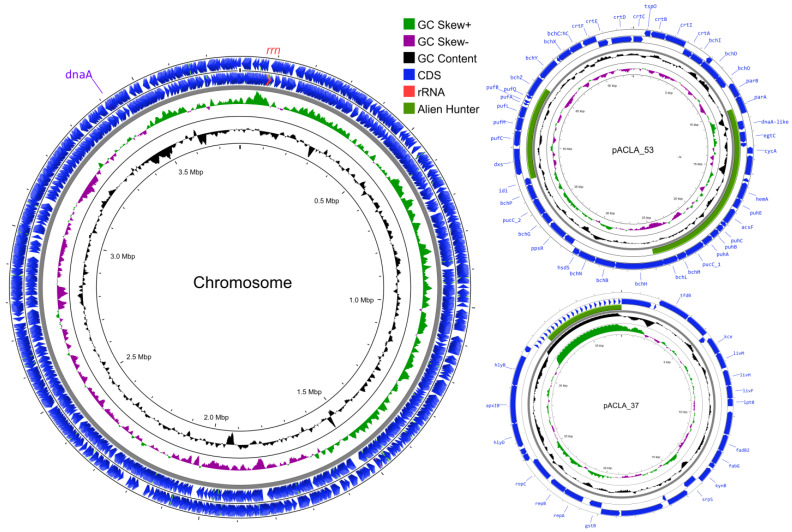
Chromosome map of 7Alg153^T^ created using Proksee server [57]. The scales are shown on the inside circles in megabases (Mbp) for chromosome and kilobases (Kbp) for plasmids. The figure also shows *rrn* operon (red label) and *oriC* (*dnaA*) (light violet label).

**Figure 4 life-15-00331-f004:**
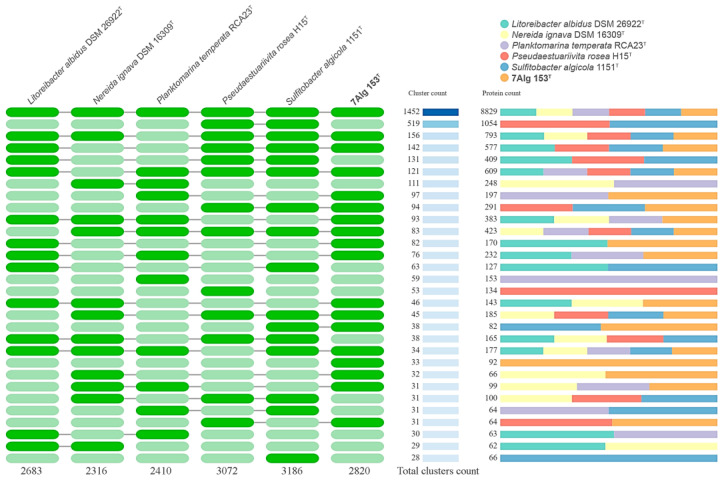
Occurrence of orthologous gene clusters among related genera strains of the 7Alg 153^T^ clade. On the left, green or light green colors indicate the presence or absence of orthologous gene clusters. The blue gradation indicates the number of orthologous gene clusters.

**Figure 5 life-15-00331-f005:**
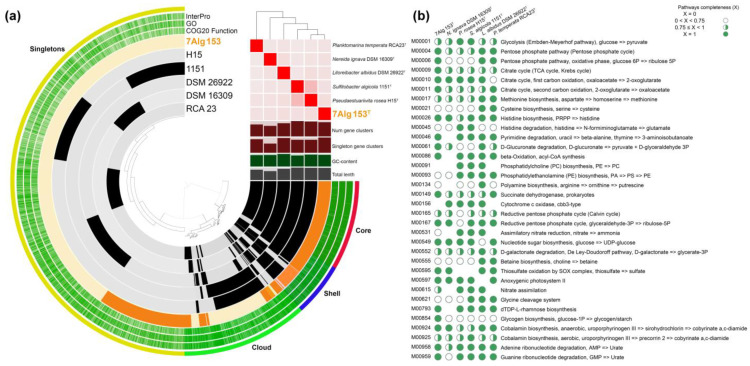
(**a**) The pan-genome of members of the 7Alg 153^T^. Circle bars show the presence/absence of 10,762 gene clusters in each genome. Gene clusters are colored as red for core, blue for shell, green for cloud, and yellow for singletons using Euclidian distance and Ward ordination. The heatmap placed in the upper right corner presents pairwise values of ANI (in %). The columns below the heatmap show the following ranges: gene cluster number (0–4093), singleton number (0–1392), GC-content (0–59%), and total genome length (0–3,966,869 bp). The strain 7Alg153^T^ is colored in orange. Other information included in the figure comprises InterPro, GO, and COG20 Function module. (**b**) Discrimination of the 7Alg 153^T^ clade members based on completeness of predicted KEGG pathway modules.

**Figure 6 life-15-00331-f006:**
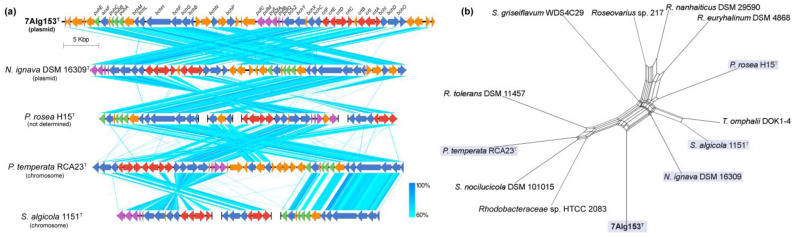
Comparison of PGC structures of 7Alg 153^T^, *Nereida ignava* DSM 16309^T^, *Planktomarina temperata* RCA23^T^, *Sulfitobacter algicola* 1151^T^, and *Pseudaestuariivita rosea* H15^T^ (**a**) and MLST neighbor-net phylogenetic network based on concatenated *bchX*, *bchY*, *bchZ*, *pufM*, *pufL*, and *puhA* translated sequences (**b**). Genes coding for photosynthetic assembly factors (*puh*) are marked with a green label, for bacteriochlorophyll biosynthesis (*bch*) with a blue label, for proteins forming the reaction centers (*puf*) with a purple label, for carotenoid biosynthesis (*crt*) with a red label, and an orange label shows genes not associated with anoxygenic photosynthetic processes.

**Table 1 life-15-00331-t001:** Genomic features of 7Alg 153^T^ and members of closely related known and potential genera of *Roseobacteraceae*.

Feature	1	2	3	4	5	6
Assembly level	chromosome	contig	contig	contig	contigs	chromosome
Genome size (bp)	3,877,416	2,837,944	3,893,358	3,966,869	3,580,729	3,288,122
Number of contigs	3	36	87	70	12	1
G+C content (mol%)	55	54	56	52	59	53.5
N50 (Kb)	3786.80	225.70	115.00	224.20	397.20	3288.12
L50	1	5	11	5	3	1
Coverage	140.0x	369.0x	100.0x	300.0x	329.0x	26.0x
Total genes	3761	2851	3983	4030	3575	3201
Protein-coding genes	3703	2753	3825	3917	3488	3108
rRNAs (5S/16S/23S)	1/1/1	1/1/1	1/1/1	1/1/1	4/4/4	2/2/2
tRNA	38	43	40	37	49	42
checkM completeness (%)	100%	99.69%	99.71%	85.66%	100%	98.58%
checkM contamination (%)	0.48%	0.92%	0.05%	7.88%	0%	0.43%
WGS project/GenBank	GCA_044772865.1	FORZ01	JACNMP01	JABUFE01	FNOI01	GCA_000738435.1
Genome assembly	ACLA_KMM_6494T_1	IMG-taxon 2599185238 annotated assembly	ASM1759233v1	ASM1331526v1	IMG-taxon 2617270827 annotated assembly	ASM73843v1

Strains: 1, 7Alg 153^T^; 2, Nereida ignava DSM 16309^T^; 3, Pseudaestuariivita rosea H15^T^; 4, Sulfitobacter algicola 1151^T^; 5, Litoreibacter albidus DSM 26922^T^; 6, Planktomarina temperata RCA23^T^.

**Table 2 life-15-00331-t002:** Phenotypic characteristics differentiating strain 7Alg 153^T^ and the type strains of the related genera of the family *Roseobacteraceae*.

Characteristics	1	2	3	4	5	6
Source of isolation	green alga *Cladophora stimpsonii*	seawater	sea snail Acmaea sp.	marine green algae	sea snail *Umbonium costatum*	seawater
Colony color	light pink	Non-pigmented	pink	light pink to light tawny	whitish	transparent to light beige
Motility	−	−	−	−	−	+
Temperature range for growth (°C):	4–32	13–28	4–37	15–37	4–37	10–30
Salinity range for growth (% NaCl):	1.5–4	1.4–8	0–7	1–6	0.5–8	1.5–5
Nitrate reduction	+	−	−	−	−	−
Acetoin production	−	ND	−	+	ND	ND
H_2_S production	−	−	−	−	+	ND
Degradation of:						
Aesculin	+	ND	−	+	+	ND
Gelatin	+	−	−	+	−	−
Tyrosine	+	−	ND	ND	+	ND
Tweens 20	+	ND	ND	−	ND	ND
Tweens 80	+	−	−	−	+	−
Urea	+	ND	+	−	−	ND
Acid formation from:						
D-Arabinose	+	ND	−	−	ND	ND
L-Arabinose	+	−	+	−	−	ND
D-Cellobiose	−	−	+	W	−	ND
D-Fructose	+	−	+	+	ND	ND
D-Fucose	−	ND	+	W	−	ND
D-Galactose	+	−	+	−	−	ND
D-Glucose	+	−	+	+	−	ND
D-Lactose	−	−	+	−	−	ND
D-Mannose	−	ND	+	−	−	ND
D-Melibiose	+	−	+	−	−	ND
D-Raffinose	−	ND	+	W	ND	ND
L-Rhamnose	+	−	+	−	−	ND
Ribose	−	−	+	+	−	ND
Sucrose	+	−	+	−	−	ND
D-Trehalose	+	−	+	−	ND	ND
D-Xylose	+	−	+	−	−	ND
Amygdalin	+	−	−	−	ND	ND
Arbuthin	+	ND	−	−	ND	ND
D-Sorbitol	+	ND	−	+	−	ND
Utilization of:						
D-Arabinose	+	ND	−	−	ND	ND
L-Arabinose	+	−	+	−	−	+
D-Fructose	+	−	+	−	ND	W
D-melibiose	+	−	+	−	+	ND
D-Glucose	+	W	+	−	−	W
Lactose	+	−	+	−	−	−
Maltose	+	+	−	−	−	W
Mannose	−	W	+	−	−	W
L-Rhamnose	+	−	+	−	−	+
Sucrose	+	−	+	−	−	ND
Amygdalin	+	−	−	−	ND	ND
N-acetyl-glucosamine	+	−	−	−	−	ND
Erythritol	+	ND	−	−	ND	ND
Glycerol	+	−	−	−	ND	ND
Sorbitol	+	−	−	+	−	ND
Mannitol	−	W	+	−	+	ND
Acetate	+	W	+	−	+	+
L-Serine	−	W	−	−	−	+
Citrate	+	W	−	−	−	−
L-Alanine, L-glutamate	+	−	+	−	−	W
L-Threonine	+	−	−	−	−	−
L-Tyrosine	+	−	ND	−	−	+
L-Histidine	+	−	−	−	−	+
Glucuronate	+	−	+	−	ND	ND
Gluconate	+	−	−	−	−	ND
Propionate	+	W	−	−	+	W
Malate	−	+	−	−	−	W
Enzyme activities (API ZYM):						
Esterase C4, esterase lipase C8	+	ND	−	+	+	ND
Naphthol-AS-BI-phosphohydrolase	+	ND	−	W	+	ND
Valine aryamidase	−	ND	−	W	−	ND
β-Galactosidase	−	ND	+	+	−	ND
DNA G+C content (mol%)	52.2	53.2	56.7	52.0	59.0	53.7

Strains: 1, 7Alg 153^T^ (this study); 2, *Nereida ignava* 2SM4^T^ [31]; 3, *Pseudaestuariivita rosea* H15^T^ [28]; 4, *Sulfitobacter algicola* 1151^T^ [13]; 5, *Litoreibacter albidus* KMM 3851^T^ [30]; 6, *Planktomarina temperata* RCA23^T^ [29]. All strains were positive for the following tests: respiratory type of metabolism and oxidase activity. All strains were negative for hydrolysis of agar and starch. +, positive; −, negative; w, weak reaction; ND, no data.

**Table 3 life-15-00331-t003:** Fatty acid profile of strain 7Alg 153^T^ and the type strains of the closely related members of the family *Roseobacteraceae*.

Fatty Acid	1	2	3	4	5	6
Saturated straight-chain						
C_12:0_	1.8					tr
C_16:0_	7.5	5.0	5.3	1.0	10.3	9.0
C_17:0_	tr				1.0	
C_18:0_	7.5	6.0	8.5	11.7	1.0	2.1
C_19:0_				1.4		
Unsaturated straight-chain						
C_12:1_					3.2	tr
C_16:1_ ω7*c*	tr					12.6
C_18:1_ ω9*c*	2.6					
C_18:1_ ω7*c*	70.2	81.4		44.1		59.7
C_18:2_					2.3	
C_19:1_					1.3	
C_20:1_ ω7*c*	tr	tr	17.6	29.7	77.1	
Branched fatty acid						
11-methyl C_18:1_ ω7*c*	5.1	3.3	4.9	6.2		11.1
Hydroxy substituted						
C_10:0_ 3-OH	1.2	1.9		1.7	2.2	
C_12:1_ 3-OH						2.3
C_18:1_ 2-OH			5.9			
Summed features *						
7		1.5		1.0		
8			46.1			

Strains: 1, 7Alg 153^T^ (this study); 2, *Nereida ignava* 2SM4^T^ [31]; 3, *Pseudaestuariivita rosea* H15^T^ [28]; 4, *Sulfitobacter algicola* 1151^T^ [13]; 5, *Litoreibacter albidus* KMM 3851^T^ [30]; 6, *Planktomarina temperata* RCA23^T^ [29]. Tr, trace (<1%); not detected; fatty acids amounting <1% in all strains are not shown; *, summed feature 7 consists of C_19:1_ ω6*c* and/or C_19:0_ cyclo ω10*c*, and summed feature 8 consists of C_18:1_ ω7*c* and/or C_18:1_ ω6*c* that could not be separated by the Microbial Identification System.

## Data Availability

The type strain of the species is strain 7Alg 153^T^ isolated from the green alga *Cladophora stimpsonii* collected in Troitsa Bay, the Sea of Japan, Russia. The GenBank accession numbers for the 16S rRNA gene and the whole-genome sequences are PQ521906 and GCA_044772865.1, respectively. Strain 7Alg 153^T^ was deposited in the Collection of Marine Microorganisms (WFCC acronym is KMM) under the number KMM 6494^T^ and in the Korean Collection for Type Cultures (KCTC) under the number KCTC 72606^T^.

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
