# Peer review of "Description and Comparative Genomics of Algirhabdus cladophorae gen. nov., sp. nov., a Novel Aerobic Anoxygenic Phototrophic Bacterial Epibiont Associated with the Green Alga Cladophora stimpsonii"

_life, 2025, doi:10.3390/life15030331_

Round 1
Reviewer 1 Report
Comments and Suggestions for Authors
Green algae of the genus Cladophora play an important ecosystem role, often mediated by epibiotic microbial representatives, which are interesting both in fundamental and practical aspects. Works in this field are relevant and necessary.
The work (life-3416383) is devoted to a comparative description of a new bacterial isolate associated with a macrophyte alga, Cladophora stimpsonii (Chlorophyta).
Comments and recommendations:
1) Please work on the title, for example, replacing «bacterium isolated from the green alga..» with «bacterial epibiont associated with the green alga Cladophora…» (and further in the text, for example, section 2.1).
2) Please include information about green algae (Chlorophyta) at the beginning of the introduction, noting that this diverse taxonomic group of algae includes both microscopic and macrophytic algae found in marine and freshwater environments.
Authors may indicate that biotechnologically significant microscopic algae of Chlorophyta include the genera Chlorella and Dunaliella as rich sources of natural value-added products, such as carotenoids (doi: 10.4014/jmb.2209.09012; doi.org/10.1080/26388081.2023.2222318; doi.org/10.3390/biology9070169), while algae of the genus Cladophora, as macroscopic green algae, are rich in pharmaceutically valuable compounds (doi.org/10.1016/j.algal.2019.101476).
3) It is also necessary to note the importance of studying the epiphytic bacterial communities of macrophyte algae.
4) LL.70-72. I strongly recommend giving at least a brief description of this important procedure.
5) Figures 1 and 2. Please indicate in the description of figures taxa used as outgroups.
6) Please improve the quality of the figures (some captions are hard to read, for example Figure 5).
Author Response
Responses to Reviewer 1.
Comment: Green algae of the genus Cladophora play an important ecosystem role, often mediated by epibiotic microbial representatives, which are interesting both in fundamental and practical aspects. Works in this field are relevant and necessary. The work (life-3416383) is devoted to a comparative description of a new bacterial isolate associated with a macrophyte alga, Cladophora stimpsonii (Chlorophyta). Comments and recommendations:
Response: Thank you very much for taking the time to review our manuscript. Thank you for your positive assessment of this study and comments as well as recommendations for improving our manuscript.
Comment 1: 1) Please work on the title, for example, replacing «bacterium isolated from the green alga..» with «bacterial epibiont associated with the green alga Cladophora…» (and further in the text, for example, section 2.1).
Response 1: In according with this recommendation, we have revised the title of our manuscript, to emphasize the bacteria-algae relationship.
Comment 2: Please include information about green algae (Chlorophyta) at the beginning of the introduction, noting that this diverse taxonomic group of algae includes both microscopic and macrophytic algae found in marine and freshwater environments. Authors may indicate that biotechnologically significant microscopic algae of Chlorophyta include the genera Chlorella and Dunaliella as rich sources of natural value-added products, such as carotenoids (doi: 10.4014/jmb.2209.09012; doi.org/10.1080/26388081.2023.2222318; doi.org/10.3390/biology9070169), while algae of the genus Cladophora, as macroscopic green algae, are rich in pharmaceutically valuable compounds (doi.org/10.1016/j.algal.2019.101476).
Response 2. Thanks for your important recommendations to extend the introduction by adding information on sources and their biotechnological potential (Lines 43-59).
Comment 3: It is also necessary to note the importance of studying the epiphytic bacterial communities of macrophyte algae.
Response 3: Since the goal of the manuscript is to identify and describe a new taxon rather than a community, we thought it would be better to say in this sentence about the need to obtain culturable bacteria (Lines 60-62).
Comment 4: LL.70-72. I strongly recommend giving at least a brief description of this important procedure.
Response 4: We added more information in Section 2.1 (Bacterial Isolation and Maintenance) (Lines 91-96).
Comment 5: Figures 1 and 2. Please indicate in the description of figures taxa used as outgroups.
Response 5: There were no outgroups for the 16S rRNA tree (Figure 1). We used the GGDC web server (http://ggdc.dsmz.de/) to estimate the phylogenetic relationships based on default parameters. The resulting tree was unrooted. Regarding the outgroup for the phylogenomic tree (Figure 2), we included this information (Lines 220-221).
Comment 6: Please improve the quality of the figures (some captions are hard to read, for example Figure 5).
Response 6: We have included high-quality Figures separately from the manuscript.

Reviewer 2 Report
Comments and Suggestions for Authors
The manuscript entitled ‘Description and comparative genomics of Algirhabdus cladophorae gen. nov., sp. nov., a novel aerobic anoxygenic phototrophic bacterium isolated from the green alga Cladophora stimpsonii' is interesting and is well-written. However, The introduction is very short. Please, bring more information about the importance (pros and cons) of bacteria to alga, its morphology. I suggest accepting the paper with minor changes.
Line 299: ‘CO2’. Please, correct to CO2
Figure 6 is not cited throughout the text.
Figure S3 is not cited throughout the text.
Author Response
Responses to Reviewer 2.
Comment 1: The manuscript entitled ‘Description and comparative genomics of Algirhabdus cladophorae gen. nov., sp. nov., a novel aerobic anoxygenic phototrophic bacterium isolated from the green alga Cladophora stimpsonii' is interesting and is well-written. However, The introduction is very short. Please, bring more information about the importance (pros and cons) of bacteria to alga, its morphology. I suggest accepting the paper with minor changes.
Response 1: Thank you very much for taking the time to review our manuscript. Thanks for your important recommendations to extend the introduction. Please, see this on Lines 43-62.
Comment 2: Line 299: ‘CO2’. Please, correct to CO2.
Response 2: Thank you. It was corrected (Line 327).
Comment 3: Figure 6 is not cited throughout the text.
Response 3: Thank you for this remark, it's fixed (Lines 300).
Comment 4: Figure S3 is not cited throughout the text.
Response 4: Figure S3 was cited in Lines 321-322 (now Lines 349-350).

Reviewer 3 Report
Comments and Suggestions for Authors
The authors isolated a new, strictly aerobic, non-motile and aerobic anoxygenic photosynthesis (AAP) bacterium from the Pacific green algae Cladophora stimpsonii, named 7Alg 153T. It was showed that the bacterium was a a new genus and a new species belonging to the family Roseobacteraceae, the order Rhodobacteales, the class Alphaproteobacteria, the phylum Pseudomonadota based on phylogenetic analysis of the 16S rRNA gene sequence. Then it was named Algirhabdus cladophorae gen. nov., sp. nov. The authors conducted a polyphasic taxonomic studies on the bacterium, and analyzed its whole genome sequence of the bacterium, and on this basis, the physiological and metabolic pathways of the bacterium were furtherly analyzed. The results are a good promotion for the study on anaerobic anoxygenic photosynthesis bacteria.
Author Response
Responses to Reviewer 3.
Comment: The authors isolated a new, strictly aerobic, non-motile and aerobic anoxygenic photosynthesis (AAP) bacterium from the Pacific green algae Cladophora stimpsonii, named 7Alg 153T. It was showed that the bacterium was a new genus and a new species belonging to the family Roseobacteraceae, the order Rhodobacteales, the class Alphaproteobacteria, the phylum Pseudomonadota based on phylogenetic analysis of the 16S rRNA gene sequence. Then it was named Algirhabdus cladophorae gen. nov., sp. nov. The authors conducted a polyphasic taxonomic studies on the bacterium, and analyzed its whole genome sequence of the bacterium, and on this basis, the physiological and metabolic pathways of the bacterium were furtherly analyzed. The results are a good promotion for the study on anaerobic anoxygenic photosynthesis bacteria.
Response: Thank you very much for taking the time to review our manuscript. Thank you for your positive assessment of this study.

Reviewer 4 Report
Comments and Suggestions for Authors
This is a relatively complete genome sequencing article, and I have a few suggestions for workers and editors to refer to.
1) The genome is not difficult now, what is the significance and necessity of the author choosing sequencing species?
2) It is necessary to not see the validation results of important genes or key functions.
Author Response
Responses to Reviewer 4.
Comment: This is a relatively complete genome sequencing article, and I have a few suggestions for workers and editors to refer to.
Response: Thank you very much for taking the time to review our manuscript.
Comment 1: The genome is not difficult now, what is the significance and necessity of the author choosing sequencing species?
Response 1: The significance and necessity of our choice to sequence 7Alg 153 genome are represented in Lines 82-86.
Comment 2: It is necessary to not see the validation results of important genes or key functions.
Response 2: Thank you for comment. The study of the key functions of some important genes of 7Alg 153 will be the focus of future research.

Round 2
Reviewer 4 Report
Comments and Suggestions for Authors
no